# Basalin is an evolutionarily unconstrained protein revealed via a conserved role in flagellum basal plate function

**Samuel Dean\*, Flavia Moreira-Leite, Keith Gull**

Sir William Dunn School of Pathology, University of Oxford, Oxford, United Kingdom

**Abstract** Most motile flagella have an axoneme that contains nine outer microtubule doublets and a central pair (CP) of microtubules. The CP coordinates the flagellar beat and defects in CP projections are associated with motility defects and human disease. The CP nucleate near a 'basal plate' at the distal end of the transition zone (TZ). Here, we show that the trypanosome TZ protein 'basalin' is essential for building the basal plate, and its loss is associated with CP nucleation defects, inefficient recruitment of CP assembly factors to the TZ, and flagellum paralysis. Guided by synteny, we identified a highly divergent basalin ortholog in the related Leishmania species. Basalins are predicted to be highly unstructured, suggesting they may act as 'hubs' facilitating many protein-protein interactions. This raises the general concept that proteins involved in cytoskeletal functions and appearing organism-specific, may have highly divergent and cryptic orthologs in other species.
DOI: https://doi.org/10.7554/eLife.42282.001

## Introduction

Flagellar motility is important for cell propulsion and the generation of fluid flow over tissues. The axonemes of most motile flagella have a central pair (CP) apparatus that is comprised of two microtubules, C1 and C2, and their projections (*Mitchell and Sale, 1999*). The CP apparatus is essential to coordinate the action of dyneins so that the flagellum beat can be propagated efficiently and defects in the CP apparatus cause cell motility defects. For example, loss of Hydin, a component of the C2 projection, causes ciliary bending and beating impairment in mice (*Lechtreck et al., 2008*), defective beat switching in *Chlamydomonas* (*Lechtreck and Witman, 2007*), and loss of locomotion in trypanosomes (*Dawe et al., 2007*). Mutations in murine Hydin are associated with hydrocephalus (*Davy and Robinson, 2003*) and human hydin is also associated with hydrocephalus (*Callen et al., 1990*) and primary ciliary dyskinesia (*Olbrich et al., 2012*). Trypanosome and *Leishmania* mutants of PF16 (a C1 projection component) are paralysed (*Branche et al., 2006*; *Beneke et al., 2017*) and PF20 (which bridges the C1 and C2 microtubules) mutants have reduced flagellar motility (*Branche et al., 2006*).

In contrast to the outer axonemal microtubules, which are extensions of the basal body microtubules, the CP does not elongate directly from the basal body. Rather, the proximal end of one or both CP microtubules is embedded in an electron dense structure at the base of the axoneme termed the basal plate (*Höög et al., 2014*; *May-Simera and Kelley, 2012*).

Although the basal plate is ideally located to nucleate the CP, the lack of basal plate mutants or known basal plate components has hampered efforts to determine its function and our knowledge of the role of the basal plate in CP assembly is limited. Ablating trypanosome gamma tubulin results in immotile flagella with no CP (*McKean et al., 2003*), suggesting that CP nucleation is mediated by the gamma tubulin ring complex. In agreement with this notion, gamma tubulin was localised to the

**\*For correspondence:**
samuel.dean@path.ox.ac.uk

**Competing interests:** The authors declare that no competing interests exist.

flagellum base in trypanosomes (*Zhou and Li, 2015*; *Scott et al., 1997*) and the TZ stellate structure in *Chlamydomonas* (*Silflow et al., 1999*). (*Silflow et al., 1999*). Knockouts of the microtubule severing protein katanin in *Chlamydomonas* and *Tetrahymena* lack the CP (*Sharma et al., 2007*; *Dymek et al., 2004*; *Dymek and Smith, 2012*), suggesting a role for microtubule severing in CP assembly. Mating katanin knockout gametes with wild type gametes caused cytoplasmic complementation and CP assembly at sub-distal regions of preformed flagellar axonemes in the resulting zygote, suggesting that CP formation does not require specific nucleating factors to be located at the base of the axoneme (*Lechtreck et al., 2013*).

Previously, we reported the development and functional analysis of a trypanosome TZ proteome (*Dean et al., 2016*). Here, we have analysed the function of a particular TZ protein that we have named basalin. Mutational analysis shows that RNAi ablation of basalin leads to paralysis and CP nucleation defects. Importantly, the TZs of cells induced for basalin RNAi possess a flagellum but are missing the basal plate, providing insights into the relationship between the basal plate and CP formation. At first sight, and intriguingly for a protein involved in such a conserved function, basalin appeared by homology searches to be a *Trypanosoma brucei* clade-specific protein. However, using syntenic comparison of the genomes we discovered a putative syntenic version of basalin in the *L. mexicana* genome. Deletion of this gene in *L. mexicana* reproduced the missing basal plate phenotype observed upon basalin RNAi in *T. brucei*, suggesting that these proteins are related but extraordinarily divergent in sequence. This establishes that proteins with intrinsic disorder are important for evolutionary conserved cytoskeletal functions and raises the general concept that such proteins involved in cytoskeletal functions and apparently appearing to be organism-specific, may have highly divergent and cryptic orthologs in other species responsible for conserved functions.

## Results

### Basalin knockdown generates flagellum paralysis, short flagella and a cytokinesis defect

Basalin (Tb927.7.3130) was identified as a putative basal plate constituent since it localised to the extreme distal TZ (*Dean et al., 2016*). We found that when basalin was ablated, the flagella of affected cells were paralysed (*Video 1*) and approximately nine microns shorter than that of uninduced cells (*Figure 1—figure supplement 1*). Cells exhibiting a supernumerary complement of DNA containing structures - the nucleus and the kinetoplast (the mass of mitochondrial DNA) - were also observed (*Figure 1—figure supplement 1*) indicative of a cytokinesis defect and concomitant reduction in population growth. There were a number of associated morphometric effects, such as a pronounced reduction in the distance between the kinetoplast and nucleus, and an increase in the length of the 'free' flagellum that extends beyond the anterior of the cell body (*Figure 1—figure supplement 1*). These defects are likely to be consequences of cell division defects caused by the shortened flagellum at cytokinesis (*Absalon et al., 2008b*; *Absalon et al., 2008a*; *Robinson et al., 1995*; *Sunter et al., 2015*; *Hayes et al., 2014*).

### Basalin ablation causes central pair defects

To determine the reason for flagellum paralysis after basalin RNAi, we repeated these RNAi induction experiments in cells expressing axonemal marker proteins tagged with a fluorescent protein. We analysed the RNAi phenotype 48 hours after RNAi induction, as this represents the first time-point after induction when basalin was undetectable at the TZ in a large proportion of induced cells, and correlates with the initial observations of morphological aberrations and growth inhibition (*Figure 1—figure supplement 1*). Two components of the CP complex (PF16 and Hydin), a component of the radial spokes

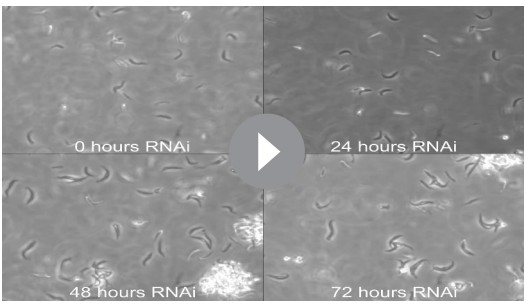

**Video 1.** An RNAi screen for paralysed flagella identifies basalin.
DOI: https://doi.org/10.7554/eLife.42282.002

(RSP2) and a component of the dynein arms were chosen as markers for this analysis. PF16 (a C1 elaboration) was absent in 63% of cells with a single flagellum after 48 hours RNAi (*Figure 1*). Dividing cells with an old flagellum that had a strong PF16 signal and a new flagellum that was negative for PF16 were often observed after RNAi, consistent with the pattern of basalin absence from the

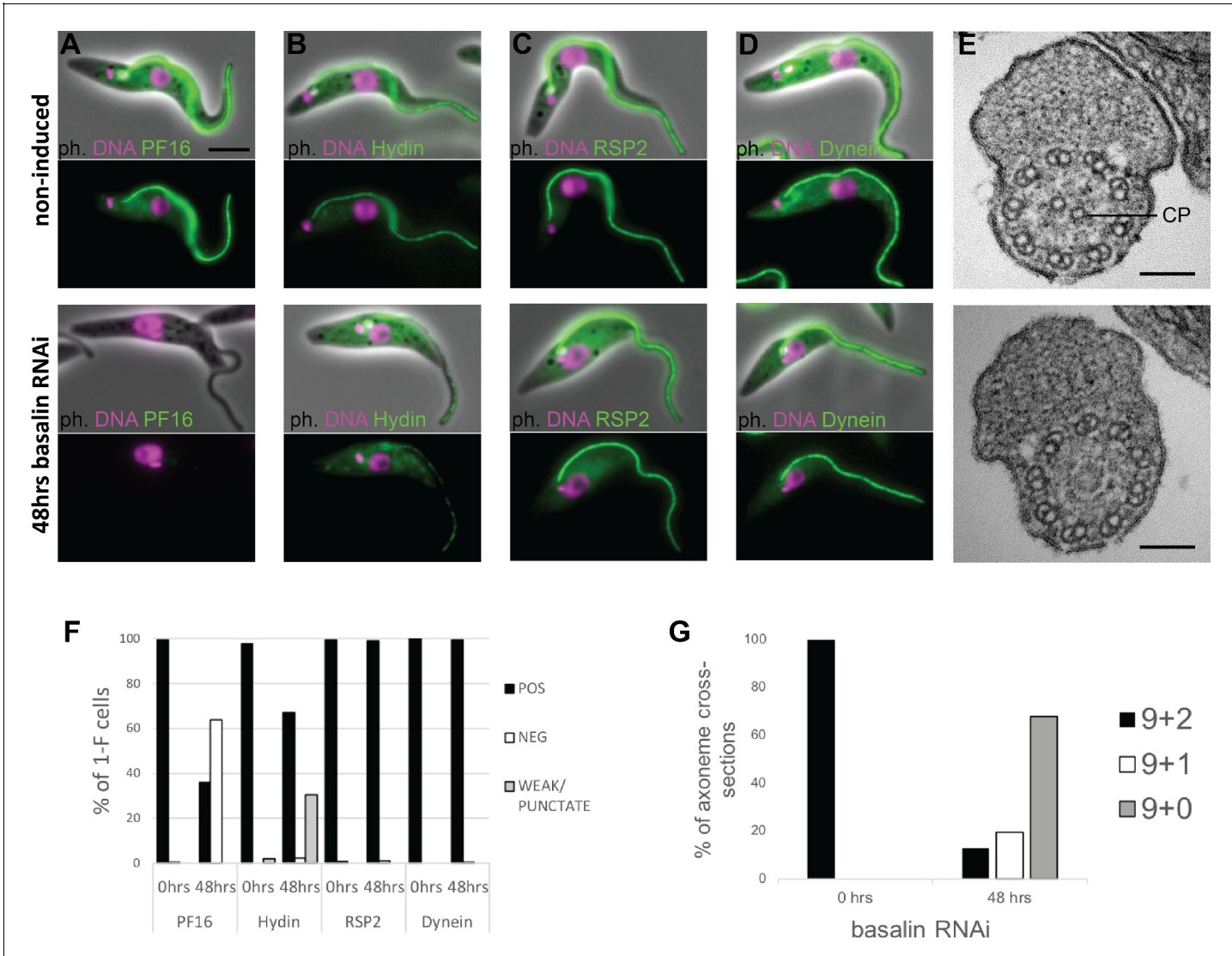

**Figure 1.** Basalin ablation causes central pair defects. Basalin RNAi was performed in cells expressing tagged markers for different axonemal components, showed that: (A) PF16 (Tb927.1.2670, C1 projection) is absent from affected flagella, (B) hydin (Tb927.6.3150, C2 projection) is reduced and punctate, but (C) RSP2 (Tb927.5.2850, radial spokes marker) and (D) dynein (Tb927.7.820, dynein arm marker) flagellar localisation are unaffected. (E) TEM cross-sections through the axoneme demonstrate that the central pair microtubules are missing from the axonemes in cells induced for basalin RNAi. (F) The presence of axoneme marker proteins (N > 200 for each marker at each timepoint) and (G) the number of axonemal microtubules were quantified before (N = 15) and after (N = 31) basalin RNAi.

DOI: https://doi.org/10.7554/eLife.42282.003

The following source data and figure supplement are available for figure 1:

**Source data 1.** Quantification of axonemal marker presence in 1F cells after 48 hours basalin RNAi.
DOI: https://doi.org/10.7554/eLife.42282.005
**Source data 2.** Quantification of axonemal microtubule arrangement after 48 hours basalin RNAi.
DOI: https://doi.org/10.7554/eLife.42282.006
**Figure supplement 1.** Quantitation of basalin knockdown and phenotype in trypanosome cells.
DOI: https://doi.org/10.7554/eLife.42282.004

new flagellum only after RNAi (*Figure 1—figure supplement 1*). Although tagged Hydin (a C2 elaboration) was still detected after basalin RNAi, its signal was fainter and more punctate in 31% of induced cells (compared with the uninduced) (*Figure 1*). Hydin remained associated with the flagellum even after detergent extraction using a non-ionic detergent, demonstrating that the residual Hydin was stably associated with the flagellum of basalin ablated cells. Flagellum localisation of RSP2 and the axonemal dynein was unaffected by basalin RNAi, suggesting that the radial spokes and dynein arms (doublet microtubules) were not significantly altered.

The flagellum paralysis and PF16 absence suggested a CP defect and hence we analysed the axonemal ultrastructure by transmission electron microscopy (TEM) of cells induced for basalin RNAi. In uninduced cells, all of the axonemal microtubules were clearly visible in all cross-sections of the axoneme, whereas in cells undergoing RNAi for 48 hours, both CP microtubules were missing in 68% of axoneme cross-sections, and 1 CP microtubule was missing in a further 20% of axoneme cross-sections (*Figure 1G*).

## Basalin ablation inhibits central pair nucleation

To gain further insights into the CP assembly defects caused by basalin ablation, we reproduced the basalin RNAi in cells expressing both basalin and PF16 tagged with different colour fluorescent proteins. This allowed us to unambiguously identify flagella with missing basalin and, by examining detergent extracted 'cytoskeletons' prepared from cells undergoing division, determine the effect upon PF16 assembly into newly made axonemes.

We then examined PF16 labelling in the new flagellum of dividing cells after 48 hours basalin RNAi, using cell cycle position defined by the number of kinetoplast and nuclei to determine the 'age' of new flagella. In basalin-negative new flagella in dividing cells, PF16 was nearly always absent at the earliest stages of elongation (97% of new flagella in cells with one kinetoplast and one nucleus, 1k1n, were missing PF16; *Figure 2*). However, at more advanced cell-cycle stages, new flagella in dividing cells were more likely to be positive for PF16 (28% and 45% of new flagella were PF16 positive in 2k1n and 2k2n cells, respectively). Strong, detergent resistant 'lines' of PF16 were observed in RNAi cells that were often in close association with the basal body, possibly reflecting self-aggregation of PF16 in the cytoplasm (*Figure 2E–H*). Close examination of the basalin-negative/PF16-positive RNAi cells revealed that PF16 incorporation was incomplete: the PF16 signal was either fragmented or missing from substantial portions of the flagellum altogether (*Figure 2F–I*). Therefore, in basalin-negative flagella, CP nucleation appears to be delayed and, when it occurs, it does so at a lower frequency and does not extend the entire length of the axoneme.

## Basalin is essential for building the TZ basal plate

To study the effect of basalin RNAi upon TZ structures, we analysed longitudinal sections of the TZ by TEM. In non-induced cells, the characteristic electron density of the basal plate was clearly observed in all longitudinal sections through the centre of the proximal region of the flagellum (comprising the basal body, the TZ and the proximal axoneme). However, after 48 hours of basalin RNAi, most longitudinal sections of the proximal flagellum either lacked a visible basal plate (57%) or had a greatly reduced basal plate only (33%) (*Figure 3*). TZ structures such as the collarette remained. The proportion of flagella lacking a basal plate correlates with the proportion of axoneme cross-sections with no CP (*Figure 1*). From this, we conclude that ablating basalin prevents the basal plate from being built and that this is associated with the absence of the CP in most axoneme cross-sections.

## Basalin localises to the basal plate area of the TZ

Basalin was previously localised to the TZ region by eYFP tagging (*Dean et al., 2016*). We tried immunogold labelling of detergent-treated cells (cytoskeletons) mounted onto TEM grids; however, we were not able to obtain reproducible gold labelling, possibly because of embedment of basalin within the basal plate. Hence, to obtain more precise localisation data on basalin, we expressed basalin fused to the bright green fluorophore 'mNeonGreen' (*Shaner et al., 2013*) in a cell line expressing mScarlet-I::TZP103.8 (*Bindels et al., 2017*) and analysed these cells by two-colour microscopy. TZP103.8 (Tb927.11.5750) was for chosen for co-localization because it was previously mapped by two-colour microscopy to a position immediately distal to the basal plate, and it is also required for CP assembly (*Dean et al., 2016*). Using this method, basalin was localised to a

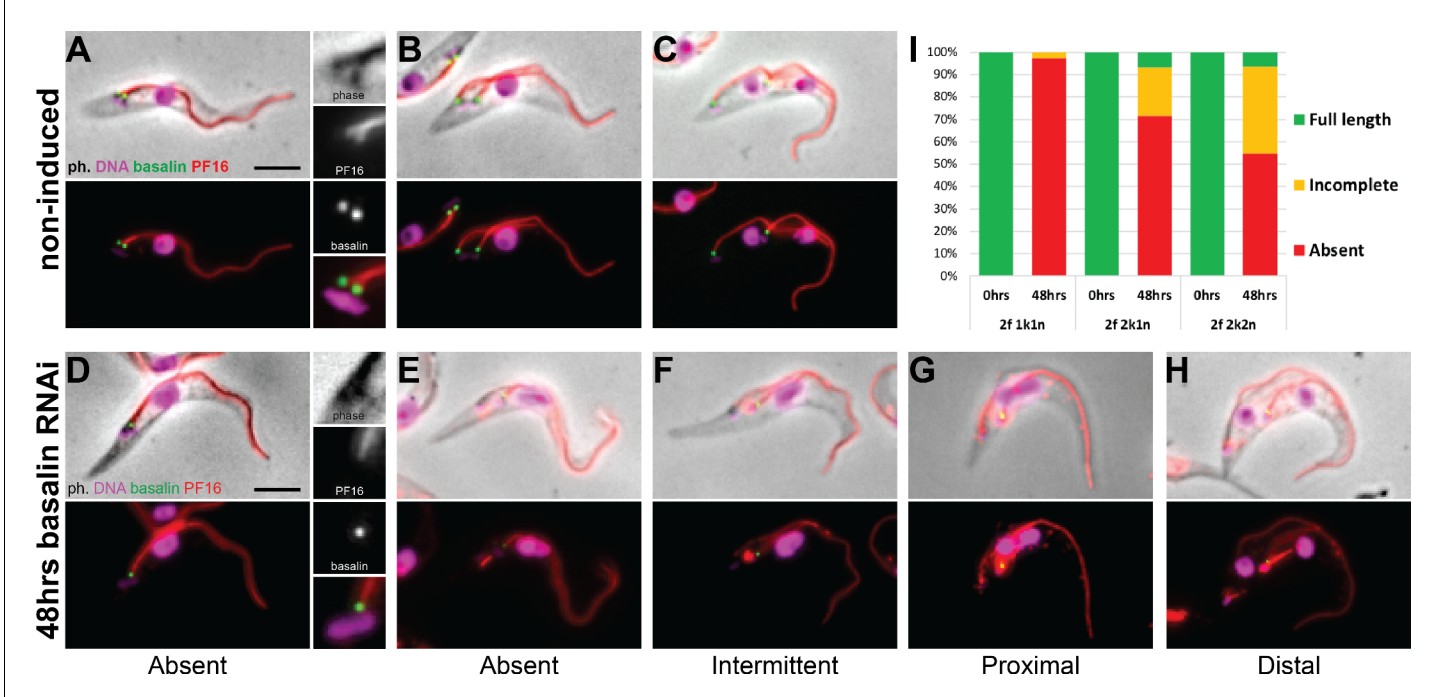

**Figure 2.** Basalin ablation inhibits central pair nucleation. Basalin was ablated in trypanosomes expressing mNG::basalin and PF16::mScarlet-I to determine the effect of basalin ablation upon newly built axonemes. (A–C) Shows non-induced cells. (A) A cell with one nucleus and one kinetoplast (1k1n) shows PF16 incorporation into the axoneme in the early stages of flagellum construction. (B) and C) show cells with 2k1n and 2k2n, respectively. (D–H) Shows cells after 48 hours basalin RNAi induction. (D) A flagellum in the very early stages of construction is missing both basalin and PF16. Older new flagella that are absent for basalin are either D) missing PF16 altogether or have incomplete PF16 assembly into the flagellum, including F) a fragmented or intermittent signal, (G) a proximal signal that does not extend to the flagellum tip, and H) a distal signal that is missing from the proximal portion of the axoneme. (I) Quantification of non-induced vs induced dividing cells at different cell-cycle positions shows that the probability of PF16 being assembled into newly-made axonemes increases with cell-cycle progression (non-induced 1k1n N = 22, 2k1n N = 50, 2k2n N = 45; induced 1k1n N = 38, 2k1n N = 60, 2k2n N = 62).

DOI: https://doi.org/10.7554/eLife.42282.007

The following source data is available for figure 2:

**Source data 1.** Quantification of axonemal PF16 incorporation in dividing cells after 48 hours basalin RNAi.
DOI: https://doi.org/10.7554/eLife.42282.008

position ~100 nm more proximal than TZP103.8 (*Figure 4*), which corresponds to the expected position of the basal plate. A faint basalin signal was also observed in the pro-basal body (*Figure 4A*), and this signal increased in intensity as the pro-basal body matured (*Figure 4B*), eventually becoming visible on both basal body/pro-basal body pairs in late dividing cells (*Figure 4C–E*). The position of basalin in the expected location of the basal plate, combined with the basal plate absence after basalin RNAi, indicate that basalin is an essential component required for basal plate formation and function.

## Basalin is essential for the efficient recruitment of TZP103.8 to the TZ

Since basalin represents the best candidate for a central component of the basal plate, we therefore next asked what affect ablation of basalin had on recruitment of other proteins which may have a function in the basal plate. Given the similarity in their phenotypes and localisations, we addressed the dependency between basalin and TZP103.8 regarding their recruitment to the TZ. For this purpose, we targeted each individual protein for RNAi separately in a cell line co-expressing fluorescently-tagged copies of both proteins. RNAi targeting basalin was effective as judged by its absence from affected TZs. Strikingly, this resulted in loss of TZP103.8 from affected TZs such that it was either undetectable or almost undetectable by fluorescence microscopy (*Figure 5B,D*). In contrast, basalin was still recruited efficiently to the flagellum when TZP103.8 was ablated (*Figure 5C,D*).

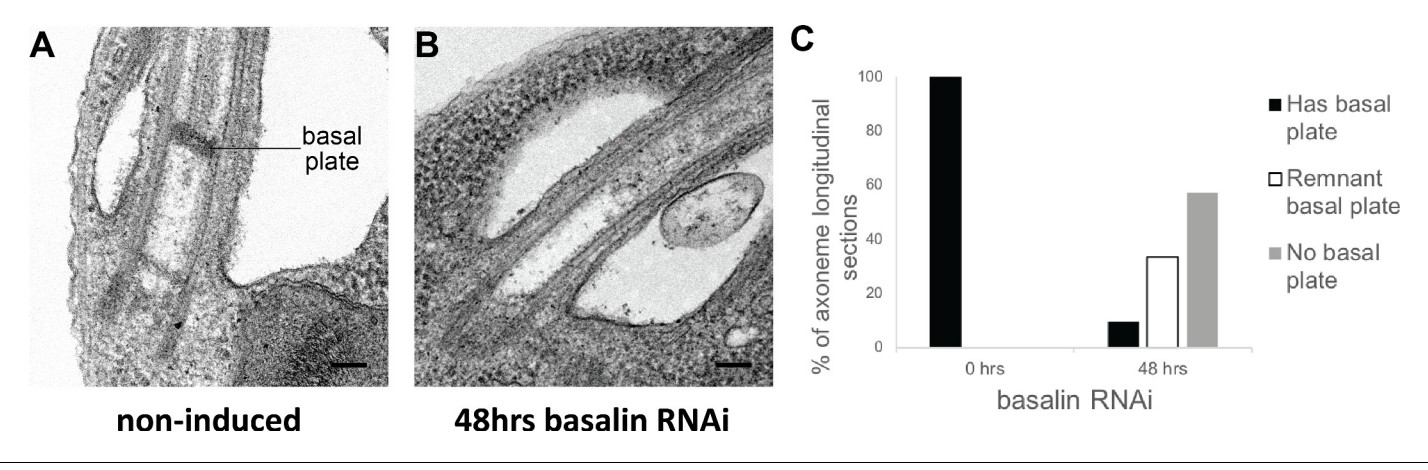

**Figure 3.** Basalin RNAi produces cells without a basal plate in the flagellum. (A) TEM longitudinal sections through the centre of the proximal flagellum (including the transition zone and the proximal end of the axoneme), before (A) and after (B) RNAi. (C) The presence and absence of a basal plate in transition zone longitudinal sections was quantified at 0 (N = 22) and 48 hours (N = 19) basalin RNAi induction.

DOI: https://doi.org/10.7554/eLife.42282.009

The following source data is available for figure 3:

**Source data 1.** Quantification of basal plate presence in TZ longitudinal sections after 48 hours basalin RNAi.
DOI: https://doi.org/10.7554/eLife.42282.010

Importantly, TZP103.8 RNAi resulted in a lack of CP microtubules but did not cause the disappearance of the basal plate (*Dean et al., 2016*), although the electron density at the basal plate was somewhat reduced in 11/32 TZ longitudinal sections ((*Dean et al., 2016*) and *Figure 5—figure supplement 1*). Basalin and TZP103.8 remained associated with isolated flagella, even after extraction with near-saturating concentrations of sodium chloride or potassium chloride (data not shown), suggesting they are core structural components of these structures.

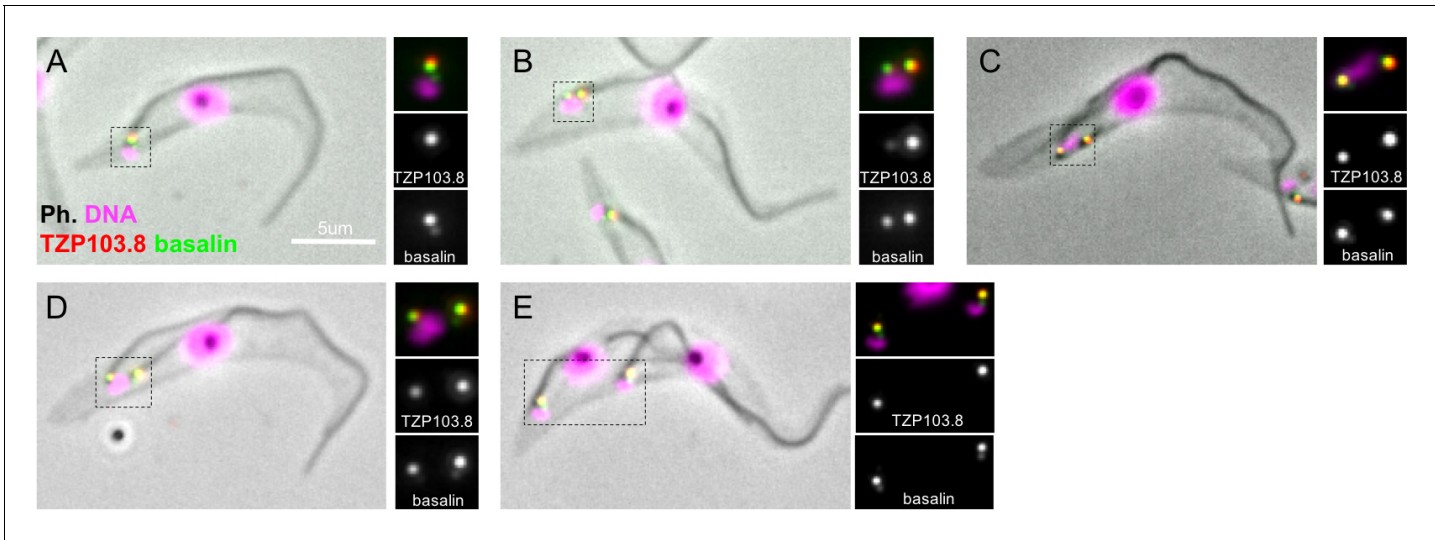

**Figure 4.** Basalin and TZP103.8 localisation through the cell cycle. (A) In non-dividing cells, basalin localises just proximal to the TZP103.8 TZ signal and faintly to the pro-basal body. (B) As the pro-basal body matures an moves to the posterior relative to the old flagellum, the basalin signal at the new transition zone increases in intensity and a faint TZP103.8 signal is detected. (C) – (E) The TZP103.8 signal intensity increases at later stages of the cell cycle, and basalin is detected on the newly formed pro-basal bodies of both old and new flagella.
DOI: https://doi.org/10.7554/eLife.42282.011

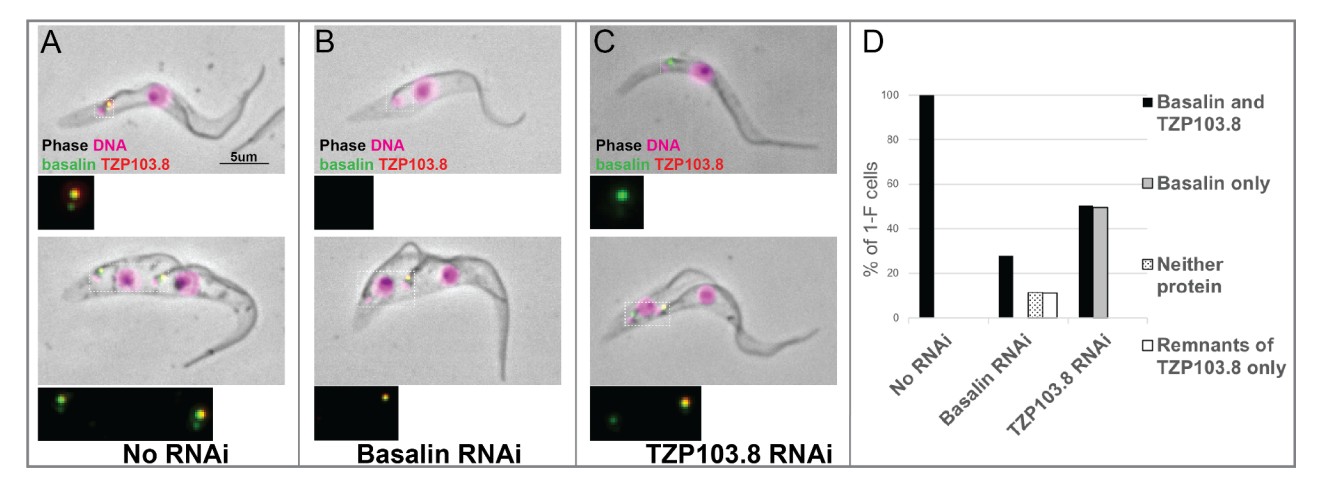

**Figure 5.** TZP103.8 depends on basalin for its localisation to the transition zone. Basalin or TZP103.8 were ablated by RNAi in cells co-expressing mScarlet::TZP103.8 (red) and mNeonGreen::basalin (green) to determine their dependency relationship at the transition zone. (**A**) In uninduced cells, basalin and TZP103.8 were both detected at the transition zone. (**B**) When basalin was ablated by RNAi, TZP103.8 was not efficiently recruited to the transition zone. (**C**) In contrast, when TZP103.8 was ablated, basalin still localised to the transition zone. Top panels show cytoskeletons prepared from non-dividing cells, bottom panels show cytoskeletons prepared from dividing cells. (**D**) Quantification of basalin/TZP103.8 presence at the transition zone in non-induced cells (N > 150) and after 48 hours RNAi of basalin (N = 168) or TZP103.8 (N = 127) confirms the differential dependencies at the transition zone.

DOI: https://doi.org/10.7554/eLife.42282.012

The following source data and figure supplement are available for figure 5:

**Source data 1.** Quantification of basalin and TZP103.8 presence at the TZ after 48 hours basalin or TZP103.8 RNAi.

DOI: https://doi.org/10.7554/eLife.42282.014

**Figure supplement 1.** Thin-section transmission electron microscopy (TEM) analysis of the basal plate after TZP103.8 RNAi.

DOI: https://doi.org/10.7554/eLife.42282.013

## Gamma tubulin basal body localisation is independent of basalin and TZP103.8 TZ localisation

Gamma tubulin has an essential role in assembling the axonemal CP (*Zhou and Li, 2015*; *McKean et al., 2003*). Therefore, we investigated whether there was a functional link between gamma-tubulin localization and the recruitment of basalin or TZP103.8 to the TZ. Firstly, we expressed gamma tubulin tagged with mNeonGreen in cells expressing either basalin or TZP103.8 tagged with mScarlet (*Figure 6*). Gamma tubulin localised ~400 nm proximal to basalin and TZP103.8, with no overlap of signal between gamma tubulin and these TZPs. This is consistent with previous data that localised components of the gamma tubulin ring complex to the basal body area in trypanosomes (*Zhou and Li, 2015*). RNAi targeting basalin caused a general reduction in the gamma tubulin signal, but did not inhibit the recruitment of gamma tubulin to the basal body (*Figure 6C and D*). Similarly, although RNAi targeting gamma tubulin resulted in the expected paralysed flagellum phenotype (data not shown), it did not prevent recruitment of basalin or TZP103.8 to the TZ (*Figure 6E and F*).

## Basalin has a highly divergent syntenic ortholog in *Leishmania mexicana*

Initial bioinformatics analyses using homology search approaches such as Orthofinder (*Emms and Kelly, 2015*) suggested that basalin had a highly restricted evolutionary distribution and was not detected outside the *Trypanosoma spp.*. These results were unexpected, given the widespread evolutionary presence of a flagellum/cilium basal plate and the fundamental and conserved nature of CP nucleation within motile cilia and flagella. Other kinetoplastid species have a motile flagellum in at least one lifecycle stage and one might therefore expect to detect a conserved protein ortholog. However, sequence analysis even failed to detect an ortholog of basalin in the *Leishmania* species, which are separated from *T. brucei* by only 300 million years (*Overath et al., 2001*; *Stevens and*

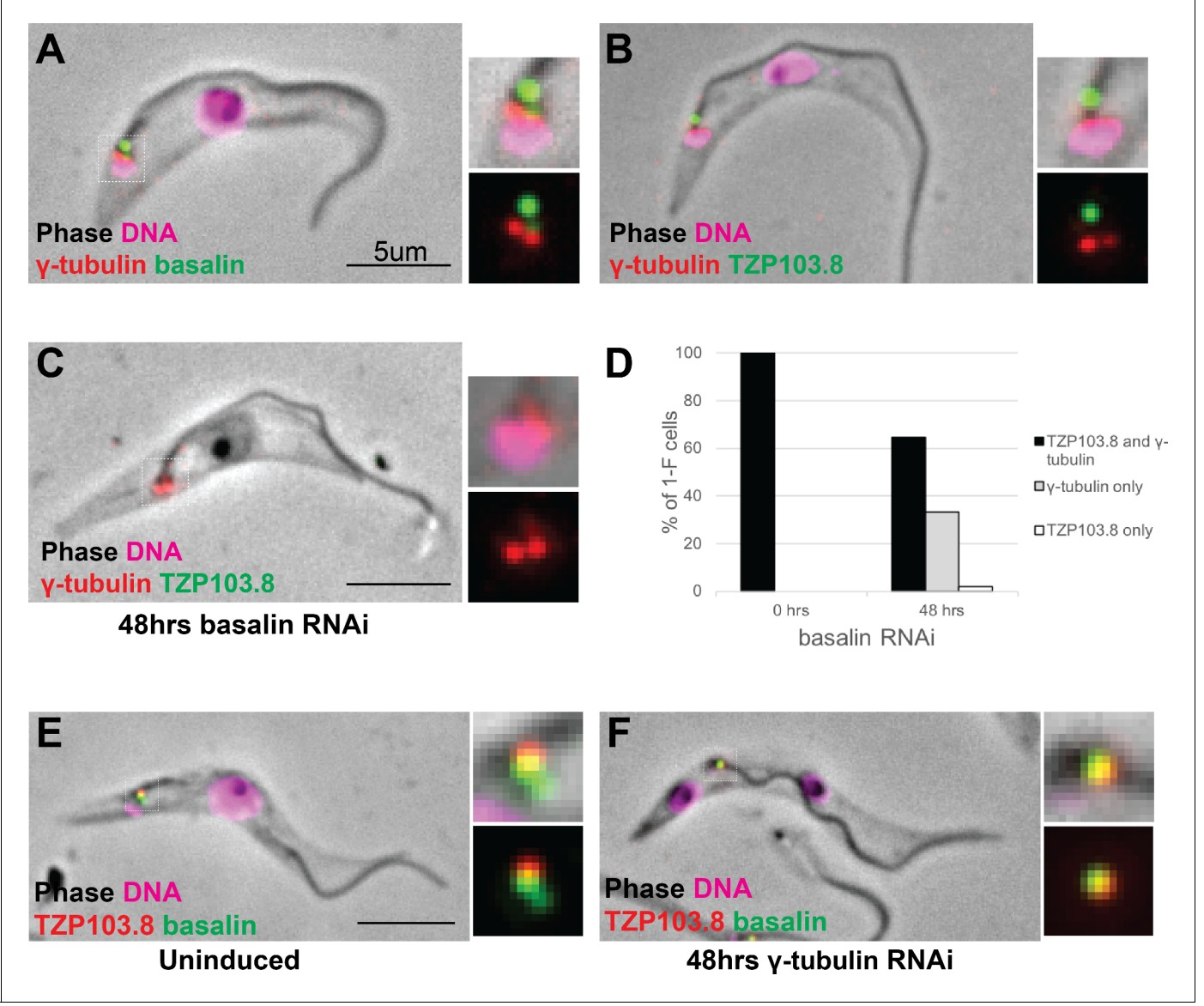

**Figure 6.** Co-localisation and dependency of gamma tubulin with basalin and TZP103.8. Gamma tubulin localises to the base of the flagellum more proximal to (**A**) basalin and (**B**) TZP103.8. (**C**) and (**D**) Ablating basalin does not prevent gamma tubulin from being recruited to the base of the flagellum (0 hr N = 107, 48 hr N = 102). (**E**) and (**F**) Ablating gamma tubulin does not prevent basalin or TZP103.8 from being recruited to the TZ (N > 100).

DOI: https://doi.org/10.7554/eLife.42282.015
The following source data is available for figure 6:

**Source data 1.** Quantification of TZP103.8 and gamma tubulin presence at the TZ in 1F cells after 48 hours basalin RNAi.
DOI: https://doi.org/10.7554/eLife.42282.016

*Gibson, 1999*; *Stevens and Rambaut, 2001*). Moreover, even in the very closely related *Trypanosoma congolense* the ortholog appeared very highly divergent to that in *T. brucei* (*Figure 7—figure supplement 1*). A striking property of most of the kinetoplastid genomes available (including that of trypanosomes and *Leishmania*) is that they are highly syntenic, which may be linked to their polycistronic transcription. Genome sequencing has revealed that kinetoplastid genes are arranged in directional gene clusters and that, even when there is extensive sequence divergence, synteny is highly conserved (*Ghedin et al., 2004*). *Leishmania spp.* are kinetoplastid parasites that can also

build a motile flagellum with an axonemal CP and a discrete basal plate in the distal TZ. Thus, we searched for a *L. mexicana* ortholog of basalin by scanning the syntenic genomic region.

Intriguingly, although gene synteny either side of the basalin gene is highly conserved between *T. brucei* and *L. mexicana*, the syntenic locus of basalin in *L. mexicana* is occupied by LmxM.22.1070, which is not a clear sequence homolog of basalin (*Figure 7*). However, we re-examined the BLASTP scores between *T. brucei* and *L. mexicana* and noticed that LmxM.22.1070 was indeed the reciprocal best BLASTP hit for basalin in *L. mexicana*, but the basalin-LmxM.22.1070 reciprocal BLASTP homologies were extremely weak (e = 0.27 and 1.9, respectively). An HHpred-driven protein comparison (*Söding, 2005*; *Zimmermann et al., 2018*) between basalin and LmxM.22.1070 demonstrated that most of the conserved residues lie in, or near, predicted helices (*Figure 7—figure supplement 1*). Although neither protein contained conserved domains that could provide functional clues, some coiled-coils and alpha helices were detected and, importantly, both proteins were predicted to be ~65% intrinsically disordered (*Figure 7—figure supplement 1*). We therefore set out to determine whether basalin and LmxM.22.1070 were divergent syntenic orthologs. To this end, we expressed LmxM.22.1070 tagged with mNeonGreen in *L. mexicana*, and demonstrated that this protein localised to the basal body and pro-basal body areas of the flagellum (*Halliday et al., 2018*) (*Figure 8*). RNAi is not possible in *L. mexicana* and so we produced LmxM.22.1070 knockouts generated using CRISPR/CAS9 (*Beneke et al., 2017*). These were viable but had substantial morphological aberrations (*Figure 9—figure supplement 1*). Similar to basalin RNAi cells, LmxM.22.1070 knockout cells had a strong motility defect (*Video 2*) slower doubling times and substantially shorter flagella, with most cells having no 'free' flagellum extending from the flagellar pocket (the cell surface invagination from which the flagellum emerges). Knockout cells were shorter and more round than the parental cells, consistent with the important link between flagellum biogenesis and cell shape (*Sunter and Gull, 2017*; *Sunter et al., 2018*).

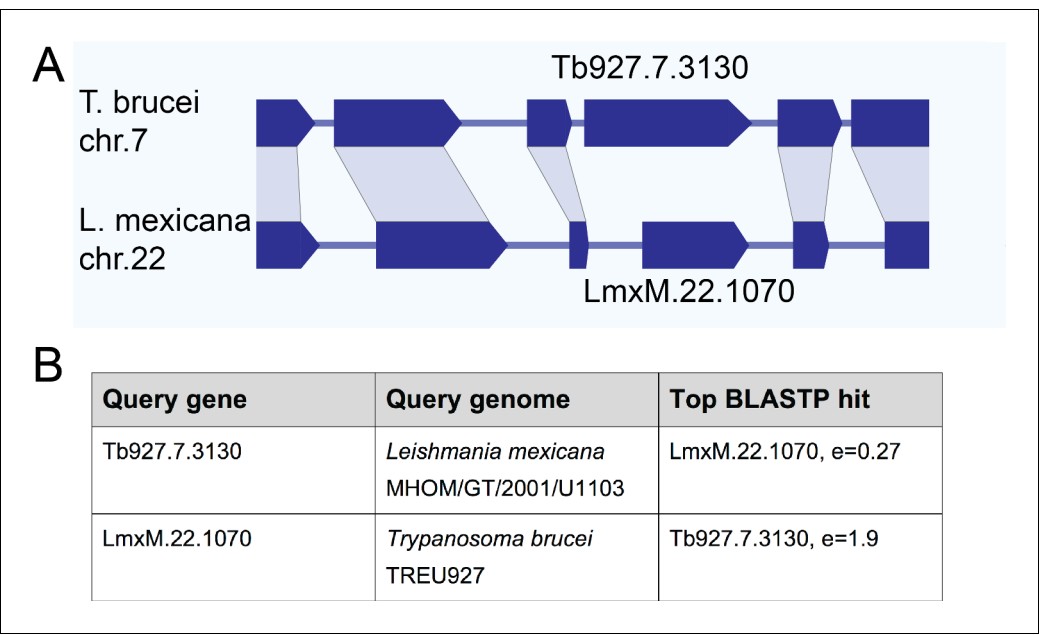

| Query gene | Query genome | Top BLASTP hit |
|---|---|---|
| Tb927.7.3130 | *Leishmania mexicana* MHOM/GT/2001/U1103 | LmxM.22.1070, e=0.27 |
| LmxM.22.1070 | *Trypanosoma brucei* TREU927 | Tb927.7.3130, e=1.9 |

**Figure 7.** Gene synteny and sequence homology between basalin (Tb927.7.3130) and LmxM.22.1070. (**A**) Basalin and LmxM.22.1070 genes are at syntenic positions along chromosome *Leishmania mexicana* chromosome 22 and *Trypanosoma brucei* chromosome 7, respectively. (**B**) Basalin and LmxM.22.1070 are very weak reciprocal best BLAST-P hits.

DOI: https://doi.org/10.7554/eLife.42282.017

The following figure supplements are available for figure 7:

**Figure supplement 1.** A bioinformatics comparison of basalin and LmxM.22.1070.
DOI: https://doi.org/10.7554/eLife.42282.018

**Figure supplement 2.** Basalin and LmxM.22.1070 are predicted to have extensive disorder.
DOI: https://doi.org/10.7554/eLife.42282.019

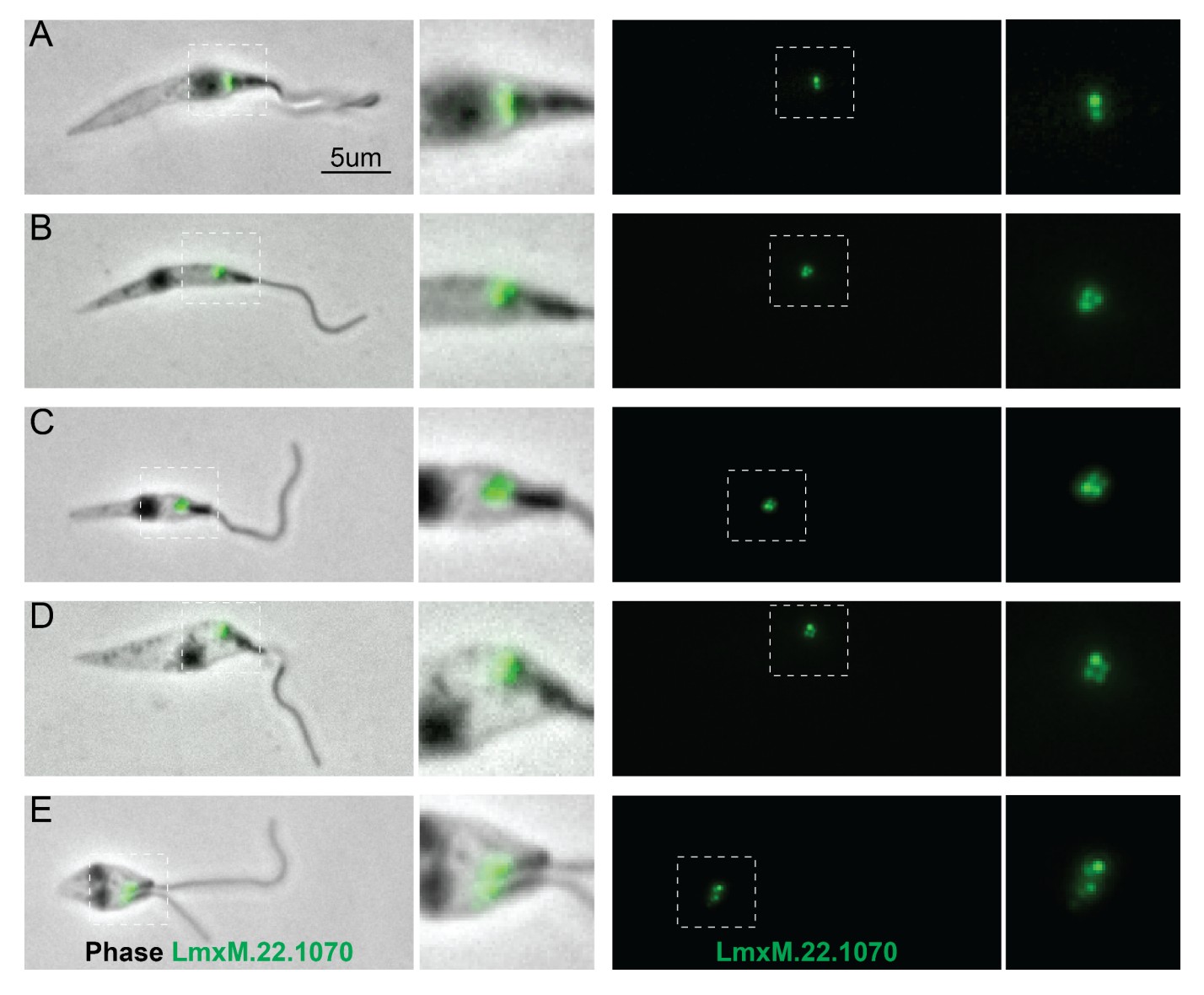

**Figure 8.** Localisation of LmxM.22.1070 in *L. mexicana* promastigotes through the cell cycle recapitulates that of basalin in trypanosomes. LmxM.22.1070 was tagged at its endogenous locus with an N terminal mNeonGreen using CRISPR/CAS9 (*Beneke et al., 2017*) and its localisation throughout the cell cycle visualised in methanol-fixed 'cytoskeletons'. (**A**) In non-dividing cells, LmxM.22.1070 localises to a position consistent with the transition zone and the pro-basal body. (**B**) - E) As the cell divides, the basal bodies are duplicated and LmxM.22.1070 is detected on the basal body and pro-basal body of both flagella in dividing cells.

DOI: https://doi.org/10.7554/eLife.42282.020

EM analysis demonstrated that, as with basalin RNAi in trypanosomes, the CP was absent from a large proportion (41%) of the axonemal cross-sections examined (*Figure 9*). Importantly, we could not detect a basal plate or an associated CP in any longitudinal sections through the TZ in LmxM.22.1070 knockout cells or in an EM tomogram (*Figure 9—figure supplement 1*), while a basal plate was nearly always observed at the TZ of parental cells (*Figure 9*). The morphological and growth defects were largely rescued by LmxM.22.1070 episome-based expression, but not by *T. brucei* basalin expression using the same system (*Figure 9—figure supplement 1*).

To evaluate PF16 incorporation in cells lacking LmxM.22.1070, we reproduced the KO in *Leishmania* cells expressing PF16::eYFP and examined cytoskeletons by fluorescence microscopy (*Figure 10*). In parental cells, PF16 exhibited the expected strong signal running from the base to the

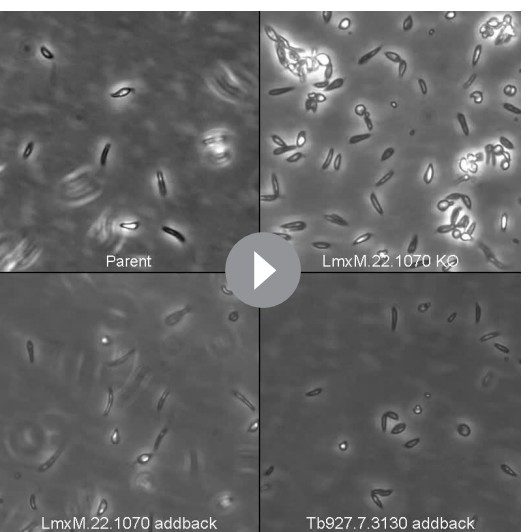

**Video 2.** LmxM.22.1070 knockout *Leishmania mexicana* cells are mostly paralysed with little or no flagellum visible.
DOI: https://doi.org/10.7554/eLife.42282.021

tip of the axoneme (*Figure 10A*). However, in those KO cells that had a clearly visible, protruding flagellum, PF16 was either completely absent (40%) or missing from a substantial portion of the flagellum (40%) (*Figure 10B–F,I*). Interestingly, KO cells with no protruding flagellum had a more penetrant phenotype, with PF16 missing from the area of the cell corresponding to the flagellum in 90% of cells (*Figure 10G,I*). These short, non-protruding flagella may be 'newer' flagella that have had less time for stochastic, ectopic assembly of PF16 and the CP. Consistent with this, cells with two 'free' flagella were not observed in KO cells, probably because the new flagellum remained short (and therefore hidden inside the flagella pocket) in most cells.

Therefore, a combination of bioinformatics, localisation and functional evidence demonstrates that LmxM.22.1070 is a highly divergent syntenic ortholog of basalin. Basalin is 'conserved' in the kinetoplastids and possibly in many other organisms where it remains cryptic because of extreme sequence diversity and lack of clues from synteny.

## Discussion

In this work, we describe basalin (Tb927.7.3130), a protein identified in our trypanosome flagellum transition zone proteome (*Dean et al., 2016*), and show that it is required for construction of the basal plate of the flagellum. Whilst a shorter axoneme is built in cells in which basalin was ablated, the absence of the basal plate leads a strong defect in CP assembly and flagellum immotility. Basalin appears to be fundamental for basal plate formation and to our knowledge this is the first example of a protein required for basal plate formation. Moreover, mutant analysis clearly demonstrates the link between the TZ basal plate structure and CP assembly. The discovery that, in basalin negative flagella, early new flagella lack PF16 and that the probability of PF16 incorporation into the axoneme increases in late new flagella, strongly argues for a defect in CP nucleation at the axoneme base, with subsequent 'ectopic' assembly in a relatively small (10–20%) of elongating axonemes. The alternative hypothesis - that the CP is nucleated correctly but loses contact with the top of the TZ and then shortens or collapses due to a stability defect - is unlikely because one would expect to see flagella in the very early stages of elongation that are positive for PF16, with decreasing frequency of PF16-positive new flagella in later cell cycle stages.

Basalin is required for recruitment of another, unrelated protein, TZP103.8, also required for CP formation. However, gamma tubulin, which again is required for CP formation, can still associate with the basal body in the absence of basalin, suggesting complex pathways are involved in CP formation. It is unlikely that the alternative view that the lack of a CP causes basal plate absence is correct since the basal plate remains in other mutants that lack the CP, such as TZP103.8 (*Figure 5—figure supplement 1* and (*Dean et al., 2016*)) and PF16 (*Beneke et al., 2017*; *Branche et al., 2006*) ablation mutants.

Although all examined TZs of *Leishmania* LmxM.22.1070 KO cells were missing their basal plate and the associated CP, 31% and 10% of axonemal cross-sections revealed a 9 + 2 and 9 + 1 axoneme, respectively. Moreover, PF16 assembly was observed in trypanosome axonemes in which TbBasalin was undetectable, indicating that some CP assembly occurs in the absence of the basal plate. How are these CPs formed in the absence of a basal plate? In TbBasalin RNAi cells, very new flagella in the early stages of construction were PF16 negative, and the probability of (often ectopic) PF16 assembly into the axoneme increased as the flagellum elongated. Further, in LmxBasalin KO cells, PF16 incorporation was much less likely in the very short, non-protruding flagella that likely correspond to the newest flagella in the population, and when PF16 was incorporation into the

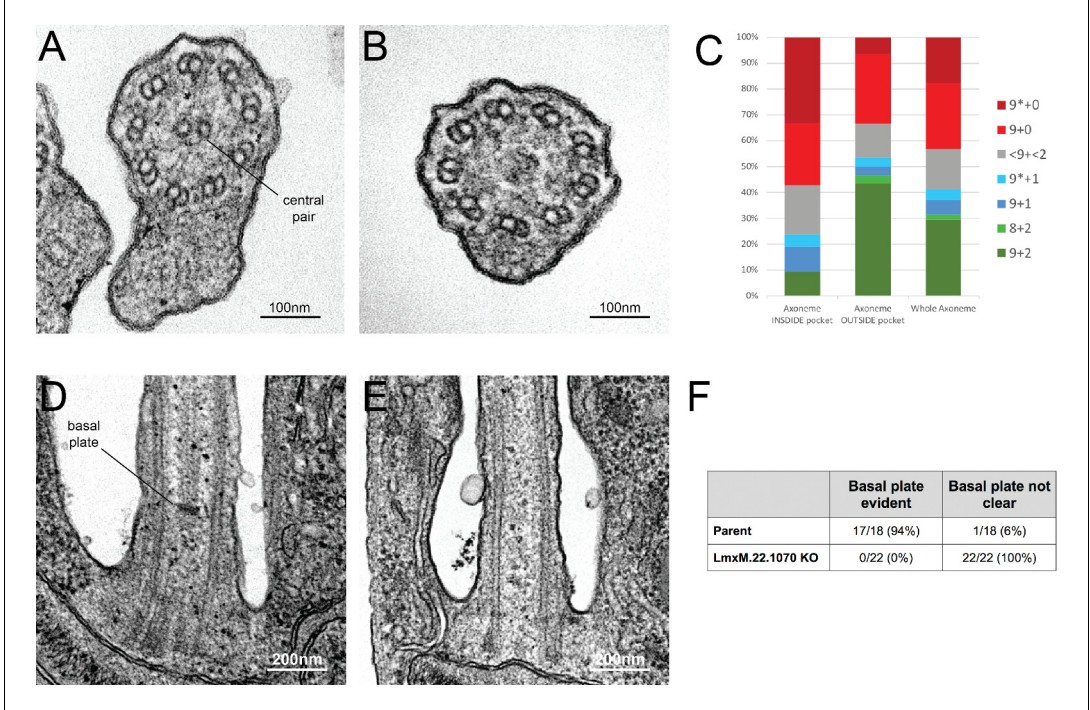

**Figure 9.** Knockout of LmxM.22.1070 in *L. mexicana* promastigotes causes flagella to be built without the central pair microtubules and a basal plate. (A) and (B) TEM cross-sections through the axoneme show that the central pair is absent in knockout cells. (C) Quantification of the microtubules arrangement in KO axonemes shows that > 40% of cross sections (N = 51) are missing their CP, and that this phenotype is more penetrant in the proximal axoneme (N = 21) versus the distal axoneme (N = 30). The asterisk refers to the microtubules being disorganised. (D) and (E) TEM longitudinal sections through the transition zone demonstrate that the basal plate is absent. (F) Quantification shows that the basal plate was not observed any the transition zones that were examined in KO cells.

DOI: https://doi.org/10.7554/eLife.42282.022

The following source data and figure supplement are available for figure 9:

**Source data 1.** Quantification of axoneme microtubule arrangement in *Leishmania mexicana* LmxM.22.1070 KO cells.

DOI: https://doi.org/10.7554/eLife.42282.024

**Figure supplement 1.** Phenotypic analysis of *Leishmania mexicana* LmxM.22.1070 knockout cells.

DOI: https://doi.org/10.7554/eLife.42282.023

protruding flagella is was usually incomplete. All of this evidence, taken together suggests that in the absence of the basal plate, CP nucleation happens at a relatively low frequency and ectopically, in the distal axoneme. Therefore, the observed CPs may reflect stochastic self-assembly of the CP microtubules inside axonemes, possibly due to mis-localisation of nucleating factors in the absence of the basal plate. Consistent with this, studies in *Chlamydomonas* demonstrate that the CP can self-assemble at different points along pre-formed flagella during cytoplasmic complementation of katanin mutants (*Lechtreck et al., 2013*). Under the normal conditions of an elongating flagellum, the presence of a basal plate-located nucleating factor may facilitate the assembly of a full CP running from the basal plate to tip.

Gamma tubulin and the gamma tubulin ring complex are essential for CP nucleation in trypanosomes (*McKean et al., 2003*; *Zhou and Li, 2015*), but the mechanism by which this occurs is not clear. Our data using fluorescent protein tagging, and previous findings using small epitope tags, localised gamma tubulin to the proximal end of the basal body, at a position ~400 nm proximal to the site of CP nucleation. A clear structural ortholog of the normal cilium TZ basal plate is not evident in the *Chlamydomonas* flagellum, but gamma tubulin has been located to the algae-specific stellate structure within its TZ (*Silflow et al., 1999*) possibly indicating a role in CP nucleation. However, in mutants where the stellate structure was lost the central pair 'penetrated' into the basal body, suggesting it may in fact function as a barrier (*Jarvik and Suhan, 1999*; *Koblenz et al., 2003*).

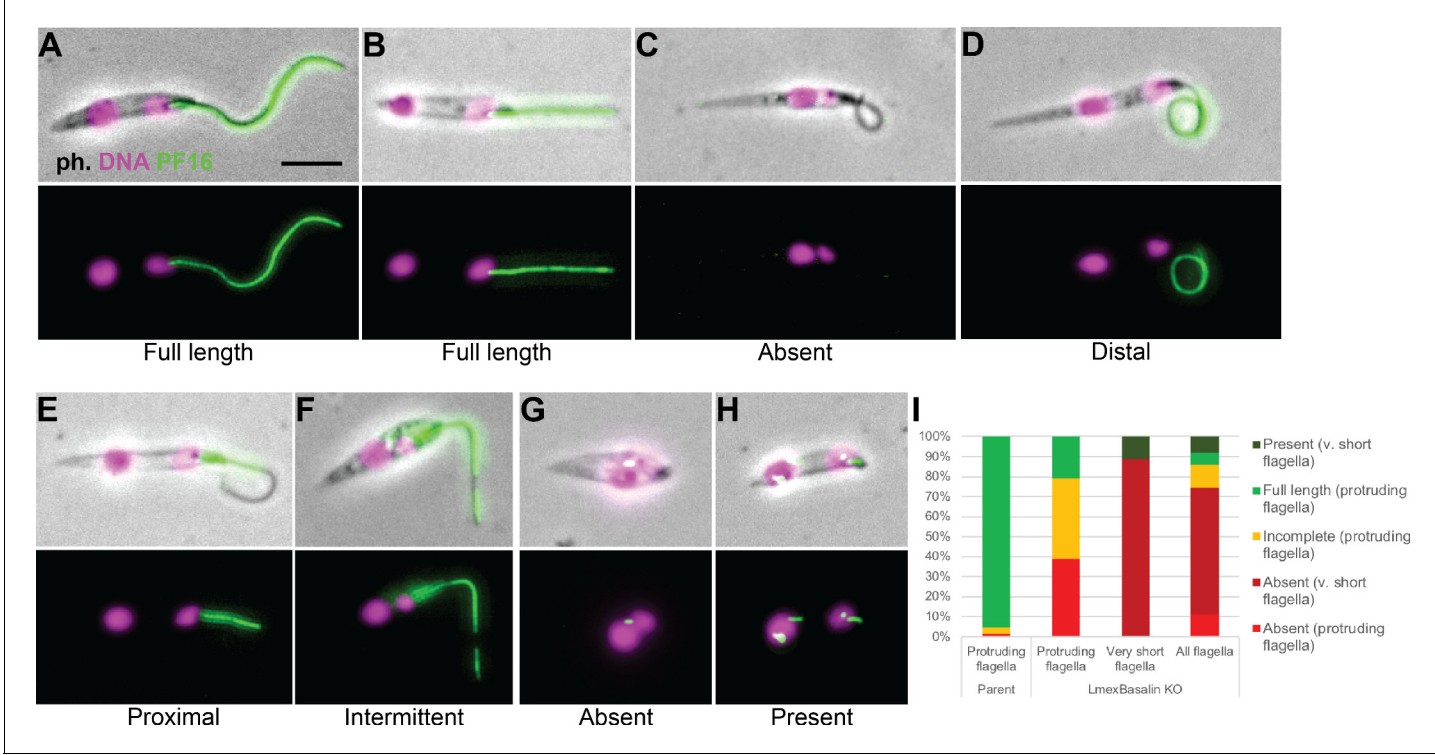

**Figure 10.** Incorporation of PF16 into LmxM.22.1070 knockout cells. LmxM.22.1070 was knocked out in *Leishmania* cells expressing a single allele of PF16 tagged with a C terminal eYFP and cytoskeletons were analysed by fluorescence microscopy. (**A**) A parental and (**B**) a KO cell with PF16 incorporated into the entire length of the axoneme. KO cells with (**C**) PF16 absent from the entire external axoneme, or (**D-F**) with incomplete PF16 incorporation. A KO cell where PF16 was either (**G**) absent or (**H**) present from the region of the cell corresponding to the proximal flagellum. Axoneme incorporation of PF16 was quantified I n (**I**) log stage cultures (parent N = 152, KO external flagellum = 62, KO no external flagellum = 152, KO total = 214).

DOI: https://doi.org/10.7554/eLife.42282.025
The following source data is available for figure 10:

**Source data 1.** Quantification of PF16 axoneme incorporation in LmxM.22.1070 KO cells.
DOI: https://doi.org/10.7554/eLife.42282.026

In contrast, the CP was not observed in the TZ in our kinetoplastid basal plate mutants, suggesting these structures are not analogous.

The concentration of gamma tubulin signal at the trypanosome basal body may represent a pre-assembly pool, possibly interacting with alpha and beta tubulins and cofactors for more distal CP assembly. Alternatively, there could be a small, undetected focus of gamma tubulin within the TZ basal plate area, although other proteins can be easily located at the basal plate. Thus, gamma tubulin may be nucleating the CP directly at the basal plate, or if the more delocalised concentration of gamma tubulin at the basal body proximal end is real then this may indicate that CP nucleation takes place at a slight distance from the basal plate. In this latter model the basal plate may act to capture a CP structure whose assembly is initiated elsewhere, rather than nucleating the CP. However, in both the nucleating and capturing models there must be number control, that is two and only two CP microtubules are incorporated into the centre of the axoneme. *Chlamydomonas* mutants that are missing radial spokes have 4 CP microtubules; similarly, axonemes from SAS6 mutants that have 10 doublets also have 4 CP microtubules, suggesting that the space available within the axoneme plays a role in limiting CP number (*Lechtreck et al., 2013*; *Mitchell, 2009*; *Nakazawa et al., 2014*). However, the origin of the extra CP in these mutants is not clear, and could be caused by a nucleation phenotype or by a single CP that breaks and slides or grows to form two parallel CPs. The absence of extra CPs in basalin mutants argues that CP number is still constrained, but does not rule out number control via specific CP nucleating or capturing sites within the basal plate.

At the base of the flagellum in *L. mexicana* parental cells, the 'lumen' of the transition zone has a different electron density from that of the inner region of the axoneme, with the basal plate delimiting a clear 'border' between the TZ and axoneme inner regions. In contrast, longitudinal sections through the centre of the proximal flagellum of basalin KO cells revealed no clear border between the inner regions of the TZ and the axoneme, as these regions had a homogeneous electron density, which may indicate a boundary function for the *Leishmania* basal plate. However, axonemal structures, such as radial spokes and dynein arms, did not penetrate into the TZ in *Leishmania* or trypanosome basal plate mutants, suggesting that the TZ/axoneme boundary remained intact despite the loss of the basal plate.

Initial analyses failed to find basalin orthologs outside the *Trypanosoma spp.* and it was only after rigorous examination of the synteny, BLAST score, localisation and function that we were able to conclude the LmxM.22.1070 is the *L. mexicana* ortholog of basalin. Analysis of the trypanosome nuclear pore complex (NPC) architecture suggested common ancestry with yeast and metazoan NPCs despite the lack of sequence conservation (*DeGrasse et al., 2009*). However, basalin divergence is far more extreme because sequence homology was not clear even within the kinetoplastids. Interestingly, both TbBasalin and LmxBasalin are predicted to contain ~65% intrinsically disordered residues. Proteins with a high degree of intrinsic disorder have been shown to associate with different elements of the cytoskeleton, including microtubules, actin, intermediate filaments and motor proteins (reviewed in *Guharoy et al., 2013*). Their roles include acting as hub proteins in protein-protein interaction networks (*Dosztányi et al., 2006*; *Dunker et al., 2005*), scaffold proteins that cause 'enforced proximity' of proteins to promote their interaction (*Buday and Tompa, 2010*), or acting as molecular 'glue' that becomes rigid once the protein complex has formed (*Uversky, 2015*). Often only a few specific peptides are conserved and these proteins acquire structure upon interaction with a binding partner (*Davey et al., 2012*; *Cumberworth et al., 2013*). Therefore, it may be that basalin is under little selective pressure to retain its primary sequence and that a few specific residues in a disordered polypeptide are sufficient to retain its interactions and function. Supporting this, the HHpred-driven alignment of TbBasalin and LmxBasalin suggests that a core helical scaffold is conserved that is rather plastic to insertions of sequences predicted as 'disordered'. Nonetheless, it is worth noting that the expression of TbBasalin in *L. mexicana* did not rescue the LmxBasalin knockout phenotype, demonstrating some degree of biochemical divergence.

Given the difficulties in identifying the relationship between the trypanosome and *Leishmania* orthologs, more distant orthologs in other organisms would not be detected using BLAST-based methods. This raises the intriguing possibility that basalin has distant orthologs in other organisms and represents a key player in a fundamental mechanism of basal plate formation and CP assembly. The discovery that a fundamental and conserved flagellum structure and function involves such widely divergent orthologues was totally dependent on our use of kinetoplastids such as *Leishmania* and trypanosomes, whose genomes have retained an enormous degree of synteny. This is one of the few groups of organisms where this type of syntenic genome analysis is possible. This raises the fundamental issue that the proportion of lineage-specific proteins in flagellar proteomes of the various model systems is likely to have been over-estimated, and that some proteins that have been categorised as clade specific may, in fact, be highly divergent ancient proteins.

# Materials and methods

## Light and electron microscopy

Live cells and detergent extracted 'cytoskeletons' were prepared for light and electron microscopy as described (*Dean et al., 2015*). All protein localisations in this study were performed by modifying genes at their chromosomal locus to encode an N or C terminal fluorescence protein tag, which was then visualised using its native fluorescence. Images were acquired using a Leica wide-field fluorescence microscope with a $63 \times 1.4$ NA objective using an sCMOS camera (Andor) and analysed using FIJI. Where the purpose was to determine the relative localisation of proteins within the cell, the microscope's chromatic aberration was measured using TetraSpec beads (Thermofisher) and then automatically corrected using a custom python script executed in FIJI (Source Code File 1). DNA was visualized by treating cells or cytoskeletons with $1 \ \mu g.mL^{-1}$ Hoechst 33342. Time lapse movies were captured using an Olympus CKX41 inverted phase-contrast microscope and a Hamamatsu

ORCA-ER camera using 100 ms exposure at maximum frame rate. Transmission electron microscopy samples were imaged in a Tecnai T12 microscope (FEI), equipped with a OneView 16-megapixel camera (Gatan).

## Cell culture and cell lines

Trypanosome SMOX 927 cells (*Poon et al., 2012*) and *L. mexicana* CAS9/T7 cells (*Beneke et al., 2017*) were cultured in SDM79 (*Brun and Schönenberger, 1979*) or M199 (*Dean et al., 2015*) for all experiments. Trypanosome and *Leishmania* cell lines were generated as described (*Dean et al., 2015*; *Beneke et al., 2017*). RNAi was induced using 2 $\mu$g.mL$^{-1}$ doxycycline. Cell densities were measured using a CASYcounter.

## Molecular cloning

Primer annealing sites for making RNAi constructs were designed by RNAit (*Redmond et al., 2003*) and the resulting amplicon was cloned into the RNAi stemloop vector pQuadra (*Inoue et al., 2005*). Primers for tagging trypanosome proteins were designed using TAGit (*Dean et al., 2015*) and primers for generating *Leishmania* knockout cell lines were designed using LeishGEdit (*Beneke et al., 2017*).

## Bioinformatics

Evolutionary analyses was initially based on Orthofinder output (*Emms and Kelly, 2015*) with subsequent refinement using BLASTP (TriTrypDB v38 with low complexity filter) and gene synteny (*Aslett et al., 2010*). Various algorithms were used to predict protein alignment (Mergealign (*Collingridge and Kelly, 2012*), HHpred (*Zimmermann et al., 2018*; *Söding, 2005*)), coiled-coils (Pair2 (*McDonnell et al., 2006*)) and intrinsic disorder (PrDOS (*Ishida and Kinoshita, 2007*)). Synteny was calculated using TriTrypDB's (*Aslett et al., 2010*; *Aurrecoechea et al., 2017*) inbuilt OrthoMCL with a 1e$^{-5}$ cut-off and the graphic reproduced using Adobe Illustrator.

## Supplemental material

*Figure 1—figure supplement 1* describes the quantitation of basalin knockdown and phenotype in trypanosome cells. *Figure 5—figure supplement 1* shows longitudinal sections of the TZ after 48 hours TZP103.8 RNAi. *Figure 7—figure supplement 1* a bioinformatics analysis comparing basalin with its *L. mexicana* ortholog. *Figure 9—figure supplement 1* is a phenotypic analysis of *Leishmania mexicana* LmxM.22.1070 knockout cells. *Video 1* shows trypanosome cells undergoing basalin RNAi. *Video 2* shows LmxM.22.1070 knockout *Leishmania mexicana* cells.

# Acknowledgements

Work in the KG laboratory is funded by the Wellcome Trust (WT066839MA and 104627/Z/14/Z). We thank Professor Matthew Higgins for his advice on basalin structure, Dr Fernando Bazan for his advice on basalin alignments, Dr Susanne Warrenfeltz and Dr Omar Harb from EuPathDB for their help on basalin synteny, and Tom Beneke for his advice on *Leishmania* CRISPR. The authors declare no competing financial interests.

# Additional information

### Funding

| Funder | Grant reference number | Author |
| --- | --- | --- |
| Wellcome Trust | WT066839MA | Keith Gull |
| Wellcome Trust | 104627/Z/14/Z | Keith Gull |

The funders had no role in study design, data collection and interpretation, or the decision to submit the work for publication.

## Author contributions
Samuel Dean, Conceptualization, Formal analysis, Validation, Investigation, Visualization, Methodology, Writing—original draft, Writing—review and editing; Flavia Moreira-Leite, Formal analysis, Investigation, Visualization, Writing—review and editing; Keith Gull, Formal analysis, Funding acquisition, Project administration, Writing—review and editing

## Author ORCIDs
Samuel Dean http://orcid.org/0000-0002-4792-2198

## Decision letter and Author response
Decision letter https://doi.org/10.7554/eLife.42282.030
Author response https://doi.org/10.7554/eLife.42282.031

## Additional files

### Supplementary files
• Source code 1. Python script to correct chromatic aberration using multi-chromatic reference beads.
DOI: https://doi.org/10.7554/eLife.42282.027
• Transparent reporting form
DOI: https://doi.org/10.7554/eLife.42282.028

### Data availability
All data generated or analysed during this study are included in the manuscript and supporting files.

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
