## [Decision Letter]

Thank you for submitting your article "Basalin: an evolutionary unconstrained protein revealed via a conserved role in basal plate function" for consideration by *eLife*. Your article has been reviewed by three peer reviewers, and the evaluation has been overseen by a Reviewing Editor and Anna Akhmanova as the Senior Editor. The reviewers have opted to remain anonymous.

The reviewers have discussed the reviews with one another and the Reviewing Editor has drafted this decision to help you prepare a revised submission.

Comments:

All three reviewers agree that the work is (1) novel, (2) of excellent technical quality and (3) supports the main conclusions. However, one major issue was raised during the discussion regarding the lack of explanation as to how loss of basalin/the basal plate affects the central pair. Some suggestions are mentioned below that could help answer this question. This would considerably strengthen your manuscript and give it a wider scope. The proposed experiments seem accessible given the tools you already have in hand.

Summary:

1) The novel protein basalin is required for the assembly of the basal plate, an electron-dense structure at the base of the central pair (CP) of axonemal microtubules. When basalin is ablated in *Trypanosoma*, the assembly of the basal plate is impaired, the central pair is mostly absent, the flagella are shorter, and cell growth/cytokinesis are affected.

2) The localization of transition zone protein TZP103.8 depends on basalin, while gamma tubulin accumulates at the basal body independently of basalin.

3) The authors identify a basalin-related protein in *Leishmania*, which displays little sequence conservation but is required for basal plate assembly, as shown by knockout experiments. The conclusion that proteins with little conservation of primary structure can perform similar functions is well supported. The fluorescence images are beautiful and very convincing. The paper is well written and structured for the most part.

Essential revisions:

How does basal plate loss affect the central pair is an important issue. We propose to perform a time course experiment during RNAi knockdown to monitor central pair presence/absence in trypanosomes. With the tools available (tagged cell lines with CP proteins, possibly TEM), it should be possible to determine whether the CP microtubules are just not made or actually lose contact with the top of the TZ and then shorten. Analysing PF16 and hydin distribution in cells with two flagella at different stages of elongation should provide essential insights. Releasing cells from basalin RNAi could also be a powerful way to monitor the relationship between CP and basal plate/basalin. The CP is still present in the *Leishmania* knockout and the authors explain this by suggesting the existence of an alternative mechanism of CP formation. It seems that this idea could be tested in *Trypanosoma*.

Another point to clarify is the differences in CP formation between *Trypanosoma* and *Leishmania* (BP-dependent vs. BP-independent). Indeed, the authors use the subtitle "Basalin is essential for building the axonemal central pair" and discuss how gamma tubulin and how the basal plate might determine the number of CP microtubules, both of which is not in agreement with published data (4 CP MTs are formed in the absence of radial spokes) nor their own data (2 CP MTs are still made in *Leishmania*, which has no BP, and gamma tub is neither at the BP nor is it distribution affected by BP loss/basalin knock-down).

The final main point to clarify relates to the role of TZP103.8. The authors report in the 2016 PNAS paper that RNAi against TZP103.8 "caused a reduced basal plate", but here in this paper cite the 2016 paper as showing RNAi against 103.8 did not cause disappearance of the BP. Could you please clarify this? For example, 103.8 RNAi causes a BP defect, albeit not "full disappearance", but retains basalin. Thus, the complete lack of BP in basalin KF, which lacks basalin + TXP103.8, could result from combined requirement of basalin and TZP103.8.

---

## [Author Response]

Essential revisions:How does basal plate loss affect the central pair is an important issue. We propose to perform a time course experiment during RNAi knockdown to monitor central pair presence/absence in trypanosomes. With the tools available (tagged cell lines with CP proteins, possibly TEM), it should be possible to determine whether the CP microtubules are just not made or actually lose contact with the top of the TZ and then shorten. Analysing PF16 and hydin distribution in cells with two flagella at different stages of elongation should provide essential insights. Releasing cells from basalin RNAi could also be a powerful way to monitor the relationship between CP and basal plate/basalin. The CP is still present in the Leishmania knockout and the authors explain this by suggesting the existence of an alternative mechanism of CP formation. It seems that this idea could be tested in Trypanosoma.

To address this issue, we made a new trypanosome cell line that targets basalin for RNAi in a cell line that expresses mNG::basalin (green) and PF16::mScarI (red). We chose to use PF16 as a CP marker because, as shown in Figure 1, this gives the clearest “absence vs. presence” pattern. This allowed us to unambiguously identify flagella that are absent for basalin and, by examining detergent extracted “cytoskeletons” prepared from cells undergoing division, examine the effect of basalin ablation upon PF16 assembly into newly-made axonemes.

Basalin-negative new flagella were not observed in a significant number of cells until 48 hours after RNAi. The delayed onset of RNAi penetrance is likely due to a cellular pool of basalin that must be depleted before basalin-negative flagella are observed. We therefore focussed on this timepoint for our analysis.

We then examined PF16 labelling in the new flagellum of dividing cells, using cell cycle position defined by the number of kinetoplasts and nuclei to score the ‘age’ of new flagella. Examining basalin-negative new flagella in dividing cells demonstrated that PF16 was nearly always absent at the earliest stages of elongation. Importantly, at more advanced cell-cycle stages, new flagella in dividing cells were more likely to be positive for PF16 (28% and 45% of new flagella were PF16 positive in 2k1n and 2k2n cells, respectively); however, flagella that were positive for PF16 but negative for basalin had incomplete PF16 incorporation, such that the central pair marker was missing from significant portions of the axoneme.

The discovery that, in basalin negative flagella, early new flagella are negative for PF16 and that the frequency of PF16 incorporation increases in late new flagella, strongly argues for a defect in central pair nucleation at the axoneme base, with subsequent ‘ectopic’ assembly in a relatively small percentage (10-20%) of elongating axonemes. The alternative hypothesis, that the central pair is nucleated correctly but loses contact with the top of the TZ and then shortens or collapses due to a stabilisation defect, is unlikely because one would expect to see flagella in the very early stages of elongation that are positive for PF16, with decreasing frequency of PF16-positive new flagella in later cell cycle stages.

Nevertheless, this new dataset does not rule out a linked/connected role for the basal plate in central pair stabilisation, in addition to its role in central pair nucleation.

The manuscript and figures have been amended to reflect this new data and the addition of this important conclusion.

Another point to clarify is the differences in CP formation between Trypanosoma and Leishmania (BP-dependent vs. BP-independent). Indeed, the authors use the subtitle "Basalin is essential for building the axonemal central pair" and discuss how gamma tubulin and how the basal plate might determine the number of CP microtubules, both of which is not in agreement with published data (4 CP MTs are formed in the absence of radial spokes) nor their own data (2 CP MTs are still made in Leishmania, which has no BP, and gamma tub is neither at the BP nor is it distribution affected by BP loss/basalin knock-down).

Our new analysis of basalin RNAi in trypanosomes expressing mNG::basalin/PF16::mScarI demonstrates that, although PF16 is mostly absent in flagella lacking basalin, PF16 can be assembled into flagellar axonemes in the absence of basalin. Importantly, this incorporation occurs at a higher frequency in “older” new flagella and is incomplete.

This strongly indicates that, in the absence of the trypanosome basal plate, central pair nucleation in these cells is delayed until after the start of axoneme elongation. This is consistent with the LmxBasalin knockout TEM data, which shows that cross-sections of the “external” (i.e., distal) axoneme of flagella outside the pocket are more likely to have a central pair than cross-sections of the proximal axoneme inside the flagellar pocket. These data suggest that delayed, ectopic central pair formation in the absence of basalin can occur in the few Basalin KO cells that manage to elongate an axoneme past the flagellar pocket opening.

To evaluate PF16 incorporation (as a surrogate for central pair formation) in cells lacking basalin, we reproduced the LmxBasalin KO in *Leishmania* cells expressing PF16::eYFP and examined PF16::eYFP incorporation in cytoskeletons. We were not able to ascertain the “age” of flagella using the cell cycle position of dividing cells because cells with two ‘free’ flagella were not observed in KO cells, likely because the new flagellum remained short and therefore hidden inside the flagellar pocket of the cell. Instead, we examined PF16 assembly into flagella that were protruding (long) and those that did not protrude from the flagellar pocket, reasoning that the long flagella were likely to be the oldest flagella.

This revealed that cells with an external flagellum were more likely to incorporate PF16, and that PF16 assembly into these flagella was incomplete and often missing from substantial portions of the axoneme. Combined with the EM data, this demonstrates that CP and PF16 assembly can occur in flagella that have no basal plate, and suggests that this assembly is often ectopic and aberrant.

In summary, our data does not support a difference in central pair nucleation mechanisms and basal plate function between trypanosomes and *Leishmania*. The manuscript and subtitle have been changed to reflect this new data.

We have modified our discussion of CP number control to include discussion of CP assembly in radial spoke mutants.

The final main point to clarify relates to the role of TZP103.8. The authors report in the 2016 PNAS paper that RNAi against TZP103.8 "caused a reduced basal plate", but here in this paper cite the 2016 paper as showing RNAi against 103.8 did not cause disappearance of the BP. Could you please clarify this? For example, 103.8 RNAi causes a BP defect, albeit not "full disappearance", but retains basalin. Thus, the complete lack of BP in basalin KF, which lacks basalin + TXP103.8, could result from combined requirement of basalin and TZP103.8.

RNAi targeting TZP103.8 causes a reduced basal plate in some sections, but most sections show little, if any change. In the PNAS paper we showed a section with a reduced basal plate, but did not quantify the proportion of affected axonemes as it was a small part of a much larger study.

To clarify this here, we have re-analysed the data to quantify the number of TZ basal plates that were affected by TZP103.8 RNAi and included the EM data as Figure 5—figure supplement 1.

The following text has been added to the Results section:

“although the electron density at the basal plate was somewhat reduced in 11/32 TZ longitudinal sections ((Dean et al., 2016) Figure 5—figure supplement 1)”

We agree that it is likely that there are proteins other than basalin that contribute to the basal plate structure that are absent from basalin mutants, consistent with our hypothesis that basalin acts as a scaffold protein.